# Predictors for E-Government Adoption of SANAD App Services Integrating UTAUT, TPB, TAM, Trust, and Perceived Risk

**DOI:** 10.3390/ijerph19148281

**Published:** 2022-07-07

**Authors:** Issam AlHadid, Evon Abu-Taieh, Rami S. Alkhawaldeh, Sufian Khwaldeh, Ra’ed Masa’deh, Khalid Kaabneh, Ala’Aldin Alrowwad

**Affiliations:** 1Information Technology Department, Faculty of Information Technology and Systems, University of Jordan, Aqaba 77110, Jordan; sf.khwaldeh@ju.edu.jo; 2Computer Information Systems Department, Faculty of Information Technology and Systems, University of Jordan, Aqaba 77110, Jordan; e.abutaieh@ju.edu.jo (E.A.-T.); r.alkhawaldeh@ju.edu.jo (R.S.A.); 3Department of Management Information Systems, School of Business, The University of Jordan, Amman 11942, Jordan; r.masadeh@ju.edu.jo; 4Faculty of Information Technology, Al-Ahliyya Amman University, Amman 19328, Jordan; k.alkaabneh@ammanu.edu.jo; 5Department of Business Management, School of Business, University of Jordan, Aqaba 77110, Jordan; a.alrowwad@ju.edu.jo

**Keywords:** unified theory of acceptance and use of technology, government, technology acceptance, theory of planned behavior, electronic government, mobile government

## Abstract

Using mobile applications in e-government for the purpose of health protection is a new idea during COVID-19 epidemic. Hence, the goal of this study is to examine the various factors that influence the use of SANAD App As a health protection tool. The factors were adopted from well-established models like UTAUT, TAM, and extended PBT. Using survey data from 442 SANAD App from Jordan, the model was empirically validated using AMOS 20 confirmatory factor analysis, structural equation modeling (SEM) and machine learning (ML) methods were performed to assess the study hypotheses. The ML methods used are ANN, SMO, the bagging reduced error pruning tree (RepTree), and random forest. The results suggested several key findings: the respondents’ performance expectancy, effort expectancy, social influence, facilitating conditions, perceived risk, trust, and perceived service quality of this digital technology were significant antecedents for their attitude to using it. The strength of these relationships is affected by the moderating variables, including age, gender, educational level, and internet experience on behavioral intention. Yet, perceived risk did not have a significant effect on attitude towards SANAD App The study adds to literature by empirically testing and theorizing the effects of SANAD App on public health protection.

## 1. Introduction

Electronic government services serve both the government and citizens alike. From a government perspective, such services save time, money, location, electricity, and water and help reduce overhead costs in general. For the citizens and country’s residents, whether tourists, workers, or investors, electronic services save time, effort, and money. Both sides of the equation will help the government and individuals. Numerous researchers tackled this issue, and the major questions are “why do people refrain from using such services?” and “how do we ease the services between government and citizens?”. In this research alone, the authors referred to more than 60 studies. Different researchers used different models to predict the factors that influence human acceptance of e-government services today. The following discusses studies conducted about e-government. In [1], 30 years ago, the researchers investigated the motivation to use computers. In published work [2] the researchers investigated online users’ perceptions of e-government. Others like [3] dug deeper to study service quality and trust issues in e-government among younger generations. the researchers sought the elderly to wear health technologies [4,5]. In [6], the researchers studied the quality of e-government websites in Pakistan, [7] London and New York, and in China [8]. In [9], the prime question was “will e-government provide the promised returns?”. In the research of [9] the authors investigated the e-government concept in two dimensions: government and citizens. The government’s areas of research are policy making, program administration, and compliance. While the citizen areas are financial, political, social, ideological, and stewardship. The two goals are enhancement of efficiency and enhancement of effectiveness. In the research [10], the researchers studied the challenges of e-government implementation in the health sector. The research of [11] found that the challenges of implementing e-government in Afghanistan fall into three major categories: organizational, social, and ICT (Information & Communication Technology) challenges”. Furthermore, Ref. [10] discussed the challenges facing e-government with a systematic review of research papers. The study found that thematic challenges fall into four categories: technological, organizational, project management, and enabling environment. The research [8] investigated the impact of e-government usage on citizen engagement during the COVID-19 crisis in China. The study showed that “e-government usage has a significant positive influence on citizens’ perception of trust in government, government transparency, and government reputation”.

The focus of this research is the SANAD App is an e-government mobile App that serves Jordanians, residents, tourists, and investors by providing a few electronic services. Services are grouped around ministries and civil services. Further, the SANAD App provides proof of vaccination and PCR results to give a pass to users to enter any public place as the law now requires. Showing a green pass allows the user to enter public places. As such, the use of the SANAD App has become mandatory as the COVID-19 virus epidemic is ravaging the world. SANAD is used by 1.642 million people. According to [12], with 48.81% saying it is complex, 15.38% saying it takes a long time to set up, 13.76% saying it takes little time, and 25.06% saying it is easy to use. A total of 7538 people took part in the questionnaire. Another application much like SANAD App is Tawakalna App used in the Kingdom of Saudi Arabia (KSA). The service of Tawakalna App is also offered to citizens and residents of KSA through Absher App [13]. Another is TousAntiCovid App which is well known in the European Union [14]. Ref. [14] used the protection motivation theory (PMT) to study the motivation to use COVID-19 protection applications. The research [15] collected data about self-quarantine from blogs to study dataveillance in South Korea. The study “explores a contact tracing mobile app for those who practice self-quarantine named the Self-quarantine Safety Protection App” The research [16] studied health risk perception, health information orientation, and perceived usefulness influence on behavioral intention, and further actual use. 

Many people consider such applications as an infringement on their privacy, while others consider such applications a form of protection from themselves and others. The question remains: What are the factors that influence people to use such an application when considering public health protection during the COVID-19 pandemic?

The objective of this research is to study the influence of using SANAD App on public health protection within the scope of discovering the different factors that influence attitude and the user’s intention to use and recommend, using well established models like UTAUT, TAM, and PBT.

The motivation for this study stems from the new idea of the SANAD App With the scope of danger such as COVID-19. As many countries are resorting to such applications within the threat of COVID-19 as stated previously. While western countries like the USA, EU do value their privacy according to laws issued in this regard, still due to necessity other countries like Jordan, Saudi and EU backed and supported such applications.

The importance of this research stems from the following: users as well as the government seek the health and well-being of the person. Furthermore, researchers, application developers, governments, and application users in general can benefit from the findings of this research. As such, both researchers and application developers can use this research as a pedestal for further research and findings and to refactor and enhance the application based on the users’ perspective of SANAD App. The Government can use the findings of this research to understand the user more and expand the satisfaction of the application user. And finally, the user of the application can use such an application with more satisfaction while enjoying the protection of such technology according to the study findings.

The empirical results offer key findings. The attitude towards SANAD App Is influenced by performance expectancy, effort expectancy, social influence, facilitating conditions, and trust. Attitude influences intent to use and intent to recommend. The intention influences public health protection.

In the light of the earlier discussion, this research relies on valid and reliable measures drawn from UTAUT, TAM, and PBT [1,17,18,19]. The research first studied if performance expectancy, effort expectancy, social influence, facilitating conditions, risk and trust, and service quality affect the attitude towards the SANAD app with intermediate variables like attitude, intention to use, and intention to recommend. Also, the total effect on public health protection. Hence the effect of using the SANAD app on public health protection. The collected data was analyzed using a structural equation modeling (SEM) approach that included a confirmatory factor analysis (CFA) that verified the constructs’ validity and reliability and machine learning (ML) methods: ANN, SMO, Bagging Reduced Error Pruning Tree (RepTree), and Random Forest. 

This research is organized in the following manner: Part 2 introduces the theoretical framework and the proposed model. in part 3, the paper describes the Survey Design and methods which includes research context, measurement items and participants and procedure. Section 4 discusses the data analysis and results; this section comprises SEM Analysis and Machine Learning (ML) Techniques results. Section 5 finally gives the research conclusions and future directions.

## 2. Theoretical Framework 

The proposed model shown in Figure 1 is based on three models: UTAUT, TAM, and TPB expanded with service quality along with four moderating factors. From UTAUT, the four major constructs adopted in the model are: performance expectancy, effort expectancy, social influence, and facilitating conditions. From TAM and TPB the three constructs: Perceived Risk, Trust in SANAD App, and Trust toward SANAD App adopted by [19], and the final construct was Perceived Service Quality [20]. The eight constructs/independent factors influence Attitude [21], which in turn influences Intention to use, which influences Intention to Recommend (CIU), which in turn influences public health protection as the model suggests. The model and hypotheses will be discussed in detail in the next two sections.

### 2.1. Unified Theory of Acceptance and Use of Technology (UTAUT)

Unified Theory of Acceptance and Use of Technology (UTAUT) introduced by [17]. UTAUT is based on the study of other models [21]. The Unified Theory of Acceptance and Use of Technology (UTAUT) model has had four major constructs since its development. The four used in this research are Performance Expectancy, Effort Expectancy, Social Influence, and Facilitating Conditions. These factors influence the attitude of a user towards the use of the SANAD App. 

Performance Expectancy (PE) is defined as: “the degree to which using a technology will provide benefits in performing certain activities” [17]. Ref. [22] stated that governments must present the advantages, benefits, and usefulness of using e-services and what sort of gains from adopting e-services. Refs. [2,22,23,24,25] researched PE, to prove that using a new technology system will aid in performance improvement, researchers found that PE has a significant impact on the individual’s Attitude to use the e-government applications and services. Therefore, we hypothesis.
**H1.** *Performance Expectancy (PE) has a positive influence related to Attitude (ATT) towards SANAD Services Adoption.*

According to [17], effort expectancy (EE) is “the degree of ease associated with the use of the system”. Several studies, such as [6,10,18] found that the easier a technology is to use, the more likely users are to adopt it, also stated that Individuals often believed that they would receive help from using user-friendly technologies. Camilleri [2] and Mensah et al. [25] stated that effort expectancy is an important predictor and has a direct and significant effect on the citizens’ intention to adopt and use e-government services. Therefore, the technological effort expectancy will decide the citizens’ intentions to adopt and use specific technologies. Previous studies found that PE influences the attitude toward use of e-government applications and services [2,19,23,25]. Consequently, H2 proposed:
**H2.** *Effort Expectancy (EE) has a positive influence related to Attitude (ATT) towards SANAD Services Adoption.*

Social Influence (SI) defined as “the degree to which an individual believes that important others believe he or she should use the system” [17]. According to [2,18], social influence is the extent to which a user’s decision to adopt and use modern technologies is an influence and inspiration by friends, close relatives, important people they respect, and others in the community. Refs. [2,18] found that SI is positively related to modern technologies. Ref. [26] claimed that SI has a moderating role related to the government e-services adoption by individuals, Kurfalı et al. [23] stated that “Social Influence measures the effect of the social environment’s attitude on the individual”, according to Mensah et al. [25] and Kurfalı et al. [23] Social influence is considered a crucial determinant when citizens make the decisions to adopt and use the e-government services.

Previous studies [2,23,25] found that there is a direct impact of SI on the individuals’ attitude to use the government e-service and application. This led to the formulation of the third hypothesis:
**H3.** *Social Influence (SI) has a positive influence related to Attitude (ATT) towards SANAD Services Adoption.*

Facilitating Conditions (FC) defined as “the degree to which an individual believes that technical and organizational infrastructure exists to support the use of the system” [17]. Furthermore, studies researched this factor-like [2,17] said that Facilitating Conditions is the Individual perception of the availability of the necessary technical and organizational ability, as well as infrastructural facilities, to enable him or her to use modern technologies effectively. In the context of e-government [17], Refs. [2,27] reported that older people need more attention when it comes to providing Facilitating Conditions such as age and experience are affected when adopting modern technologies. Several studies have shown that Facilitating conditions is the extent to which citizens believe there are sufficient resources to encourage and support them in using and having access to e-government services [23,28,29]. Previous studies [23,25,28] found Facilitating Conditions has a major influence on an individual’s attitude towards using e-government applications and services. Thus, this led in this research to the following hypothesis:
**H4.** *Facilitating Conditions (FC) has a positive influence on Attitude (ATT) towards SANAD Services Adoption.*

### 2.2. Technology Acceptance Model (TAM) & Theory of Planned Behavior (TPB)

The first modified TAM was developed in 1989 according to [21,30]. Meanwhile, the final version of TAM was developed in 1996. Attitude influences intention, which in turn is related to the actual usage. according to [30] perceived usefulness and ease of use influenced attitude towards use. On the other hand, the Theory of Planned Behavior (TPB) was developed in 1991 [21] and has an attitude that influences intention, which in turn influences behavior. According to [30], TPB has two versions: the original and a decomposed TPB.

Ref. [31] stated that citizens believe that there is a fair amount of loss when adopting and using government e-services and its applications that require using and accessing the internet. According to the previous studies [3,19,25,32], all found that perceived risk (PR) might limit the adoption of the interaction of citizens with the government e-services and applications. As a result, citizens might have a negative view impacting their attitude (ATT) toward reusing government e-services and applications. The following hypothesis is drawn from the above literature review:
**H5.** *Perceived Risk (PR) has a negative influence related to Attitude (ATT) towards SANAD Services Adoption.*

Trust and Perceived Risk were investigated by [3,19] as part of the technology acceptance model (TAM) and theory of planned behavior (TPB). The research [19] states that some studies linked trust to TAM in the realm of e-business, others linked trust to TPB in the realm of e-government, and fewer studies integrated trust and PR to TAM and TPB in an e-government context. Hence, this study integrated not only trust and PR but also performance expectancy (PE), effort expectancy (EE), social influence (SI), and facilitating conditions (FC). Previous studies conducted by [25,33] indicated that government e-services adoption is based on a high level of trust where trust is considered an important determent. Kurfalı et al. [19], Bélanger and Carter [32] and Xie et al. [25] argue that the most critical aspect of e-government services and applications adoption is trust. Al-Gahtani [34] stated that “trust of e-government is allowing individuals to willingly use e-government services and behave in a socially responsible manner for the fulfilment of trust after taking government characteristics into consideration.” Accordingly, trust towards e-government services is the citizens’ perceptions of e-government services’ competence and integrity to provide expected services. Moreover, [3] found that trust in the government has the strongest direct influence on citizen loyalty to government e-services adoption. There are two aspects of trust, [19] trust toward the e-government, while in [3,32] trust in e-government. However, most studies validated that trust affects individuals’ attitudes, it is found that trust increases the adoption level of governmental e-services and applications [3,19,25]. Hence, and based on the preceding, the following hypotheses are formulated:
**H6.** *Trust towards SANAD App (TT) has a positive influence related to Attitude (ATT) towards SANAD App services Adoption.*
**H7.** *Trust in SANAD App (TI) has a positive influence related to Attitude (ATT) towards SANAD Services Adoption.*

According to [35] perceived service quality is “the outcome of an evaluation process where the customers compare their expectations with service they have received”. While [36] said that perceived service quality is considered as the difference between expected services, and perceptions of the actual service delivered are described as perceived service quality. Perceived service quality defined as the variation between the expected services and perceptions of the actual service provided. According to [7,20,37,38] service quality has many standards that can be grouped into 5 dimensions “tangibles (SQT), reliability (SQR), responsiveness (SQP), assurance (SQA), and empathy (SQE)”. Previous research has shown that Perceived Service Quality (PSQ) has a noteworthy influence on behavioral intentions to use e-government services and to recommend the adoption of e-government services [3,25,39]. In addition, researchers found that the quality of service has the greatest influence on citizen loyalty. Hence, and based on the previously mentioned research the following hypothesis formulated:
**H8.** *Perceived Service Quality (PSQ) has a positive influence related to Attitude (ATT) towards SANAD Services Adoption.*

The First modified version of the Technology Acceptance Model (TAM) [40] suggested that Attitude (ATT) influences the Intention to Use (INU). The same model, TAM, also suggested that Intention Use (INU) influences the Intention to Recommend or the Actual Use. Researchers [25,31] found that the degree to which an individual user expresses a positive or negative assessment of engaging or interacting with technologies, such as e-government services, is referred to as their attitude toward technology adoption. Ref. [19] reported that the attitude of users has been influenced by perceived usefulness, perceived ease of use, trust toward government e-services, and perceived risk. Moreover [19,29] have shown that attitude has a direct impact on behavioral intentions to use government e-services. So, the following hypothesis formulated.
**H9.** *Attitude (ATT) has a positive influence on Intention to Use (INU) SANAD Services.*

The Technology Acceptance Model (TAM) [40] has shown that the intention to use the modern technology is determined by the individuals’ expectation about the usefulness and ease of use. Also, UTAUT model studies found that the intention to use and recommend modern technology is affected by the individuals’ performance expectancy, social influences, and effort expectancy in addition to the facilitating conditions [18,19,25]. Refs. [41,42,43] studies conducted to investigate Citizens’ intention to use and recommend e-participation using the UTAUT model. Researches [44,45] investigated the intention to recommend the e-government services in terms of mobile payment adoption, based on the previous studies, researchers found that users who intend to use the governmental e-services and its application, have a positive influence on intention to recommend its adoption [2,19,29,44]. Accordingly, the following hypothesis formulated:
**H10.** *Intention to Use (INU) SANAD Services has a positive influence on Intention to Recommend (IRC) SANAD Services.*

The public health protection construct studied by [28], in the context of the effect of social networks and their effect on public health protection by increasing awareness through behavioral change. Refs. [4,5] investigated the Continued use intention of wearable health technologies among the elderly. Researches like [28] found that modern technologies can positively affect awareness and increase public protection against the COVID-19. Based on the previously mentioned literature review the following hypothesis is formulated.
**H11.** *Intention to Recommend (IRC) SANAD App has a positive influence on public health Protection (PHP) from COVID-19.*

### 2.3. Moderating Factors Hypothesis

In addition to eight independent factors, the study suggested 4 moderating factors that influence the intermediate variable attitude towards SAND App The moderating factors suggested are age, education level, gender, and internet experience. The moderating factors were developed based on many previous studies and suggested in the UTAUT original model [7,28,46].

Gender as a moderating factor affecting the intermediate factor attitude was suggested in the original UTAUT model as a moderator [18,28,45,46,47,48,49]. Based on the previous research the following hypothesis was developed.
**H12.** *Gender has a significant moderating effect on Attitude (ATT).*

Age is an important moderating factor with two contradictory points of view. The first point of view advocates that older mobile App users are hindered by eyesight problems and are less keen to use mobile applications i.e., SANAD App On the other hand, older people are more responsible and will use SANAD applications due to greater good since it is intended to protect their health. Many studies conducted the research about age as a moderator factor influencing intermediate variable attitude i.e., [5,18,46,47,50,51]. Hence the following hypothesis was developed.
**H13.** *Age has a significant moderating effect on Attitude.*

Educational level as a moderator factor that will influence attitude towards SANAD App was adopted by [47,52]. Educational level, intuitively, is an important moderating factor. With the belief that education level influences the attitude of a user positively. Based on previous the following hypothesis was developed
**H14.** *Educational level has a significant moderating effect on Attitude (ATT).*

As said by [47,48], internet experience shows the amount of time that a user spends using certain technologies i.e., e-commerce, m-banks, etc. Hence, the following hypothesis was developed:
**H15.** *Lack of experience in use of the Internet has a significant moderating effect on Attitude toward SANAD App.*

To support the literature of this study, the researchers depended on current published research that pertains to e-government services and quality almost referenced 26 papers on this topic. Each hypothesis was referenced by at least 5-6 current published papers. Each used model was referenced by UTAUT 17 studies, TAM 9 studies, PBT 4 studies.

## 3. Survey Design/Methods

This research paper aims to investigate the overall impact on public health protection (PHP) of the SANAD App, a government’s digital and mobile services in Jordan. The research looks at effort-easing conditions, expectancy, social influence, perceived risk, performance expectancy, perceived service quality trust, and the effect on user attitude. Also investigates user attitudes toward intent to use, intent to recommend the SANAD App, and the impact on public health protection (PHP). Because research on this topic was limited, the researchers developed a suggested model in Figure 1, which led to the development of the hypothesis above. Furthermore, a questionnaire was developed and evaluated, and data was collected from 442 contributors using a convenience sample. 

### 3.1. Research Context

The world’s public health protection is the most important currently as COVID-19 attacks people. All tools and techniques are used to protect our loved ones against such deadly viruses. In this research, using SANAD App will help protect public health, and what are the factors that will influence our attitude to use and recommend using such applications. Hence, this study was conducted as follows. 

### 3.2. Measurement Items

To evaluate the proposed research model for this study, a questionnaire developed. The questionnaire items developed based on previous studies. There are 14 variables in the model. Attitude (ATT) and Intention to Use (INU) were each measured by 3 items derived from [19] and have been slightly altered to fit this research. Intention to Recommend (IRC) was measured by 3 items derived from [53,54,55,56] and has been slightly changed to fit this study. Public Health Protection (PHP) was measured by 3 items derived from [28] and has been slightly changed to fit this study. Trust towards E-Government (TT) was measured by 3 items derived from [19] and has been slightly changed to fit this study. Trust in E-Government (TI) was measured by 3 items derived from [3,32] has been slightly changed to fit this study. Perceived Risk (PR) was measured by 3 items derived from [19] and has been slightly changed to fit this study. Service Quality (SQ) has 5 aspects: Tangible (SQT), reliability (SQR), responsiveness (SQP), assurance (SQA), and empathy (SQE). Tangible (SQT) was measured by 4 items, SQP was measured by 4 items, while SQA and SQE were each measured by 3 items derived from [3]. Effort Expectancy (EE) was measured by 4 derived from [57]. The constructs PE, SI, and FC were measured by 3 items each derived from [57]. Table A1 summarizes the constructs and their measures for each original source. 

### 3.3. Participants and Procedure

A web-based Google Docs survey questionnaire in Arabic and English was created, with a 5-point Likert scale ranging from strongly disagree (1) to strongly agree (5). A panel of 11 academics reviewed the survey. The questionnaire was revised in response to the feedback.

Consequently, the survey was piloted on 25 SANAD App users in Jordan to test the understandability of the questions. Revisions were made to the survey during 15 December 2021 to 4 January 2022. The survey was performed on a sample of convenience of 442 SANAD App users living in Jordan and the respondents are reflected in Table 1 (demography). As shown in Table 1, the demographic profile of the respondents for this study showed that they are typically females’ gender, between 18-less and 34 years of age, the majority held bachelor’s degrees, and had excellent internet experience. 

## 4. Data Analysis and Results

The data analysis for this research included the first descriptive analysis to measure respondents’ attitudes. Second, Structural Equation Model (SEM) included: First, Confirmatory Factor Analysis (CFA) was conducted. Next, Structural Equation Modelling (SEM) using Amos 20 was made to evaluate the study hypotheses. Third, the Moderating effects. Finally, using artificial intelligence methods to confirm and predict public health protection using SANAD App.

### 4.1. Descriptive Analysis 

The mean and standard deviation were estimated to illustrate the responses and thus the respondents’ viewpoint on each question asked in the survey. While the mean represents the data’s central tendency, the standard deviation measures dispersion and provides an index of the data’s spread or variability [58,59]. In other words, a small standard deviation for a set of values indicates that these values are clustered closely around or close to the mean; a large standard deviation indicates the opposite. The level of each item was determined by the following formula: (highest point in the Likert scale—the lowest point on the Likert scale)/the number of the levels used = (5 − 1)/5 = 0.80, where 1-1.80 reflected by “very low”, 1.81–2.60 reflected by “low”, 2.61–3.40 reflected by “moderate”, 3.41–4.20 reflected by “high”, and 4.21–5 reflected by “very high”. Then the items were ordered based on their means. Table 2 and Table 3 show the results. 

As given in Table 2, data analysis results have shown that all research variables are applied to moderate levels, while respondent’s attributes of FC and EE do exist highly with a mean of 3.5830 and 3.4661, respectively. Among the mediating factors, IRC was the highest-ranking factor. Table 3 shows the mean, standard deviation, level, and order scores for items to each variable. 

### 4.2. SEM Analysis 

In this study SEM analysis was utilized to test the research hypotheses in two steps. First, Confirmatory Factor Analysis (CFA) was performed. Next, Structural Equation Modelling (SEM) using Amos 20 was made to test the study hypotheses.

#### 4.2.1. Measurement Model 

Confirmatory factor analysis (CFA) was used to validate the instrument items’ properties. Indeed, the measurement model demonstrates how latent variables or hypothetical constructs are evaluated in terms of observed variables, as well as the validity and reliability of the observed variables’ responses to the latent variables. [60,61,62,63]. Table 4 shows the factor loadings, Cronbach alpha, composite reliability, and Average Variance Extracted (AVE) for the variables. All the indicators of the factor loadings exceeded 0.50, except one item (AT2 = 0.258) was eliminated to obtain a better fitting measurement model, thus constituting evidence of convergent validity [60,64].

Indeed, while the measurement reached convergent validity at the item level because all the factor loadings went above 0.50, all the composite reliability values exceeded 0.60, showing a high level of internal consistency for the latent variables. In addition, since each value of AVE exceeded 0.50 [60,61], the convergent validity was proved.

Also, as observed in Table 5, all the intercorrelations between pairs of constructs were less than the square root of the AVE estimates of the two constructs, providing discriminant validity [61]. Accordingly, the measurement results show that this study had adequate levels of convergent and discriminant validity.

Further, one can draw from Table 5 that Perceived Risk has the lowest correlation with respect to the other variables, the correlation is below 0.7. Also, low correlation between Facilitating Conditions, and all the variables except Effort Expectancy. On the other hand, high correlation (above 0.95) between Trust in SANAD App and Trust towards SANAD App, Attitude, Intention to Use and Intention to Recommend, and Intention to Recommend. High correlation (above 0.93) between Performance Expectancy and both Social Influence and Perceived Service Quality. Next, High correlation (above 0.90) between Trust in SANAD App and Perceived Service Quality, Public Health Protection and Trust in SANAD App Trust towards SANAD App, and Intention to Recommend. There is a correlation above 0.80 between PE and EE, TI, TT, ATT, INU, IRC, and PHP. Also, between EE and SI, TI, TT, and PSQ. In addition, between SI and TI, TT, PSQ, ATT, INU, IRC, and PHP. Similarly, between TI and ATT, INU, IRC. Likewise, between TT and PSQ, ATT, INU, and IRC. Comparably between PSQ and ATT, INU, IRC, PHP. Lastly, between ATT and PHP, INU, and PHP.

#### 4.2.2. Structural Model

Structural equation modelling using Amos 20 was performed to test the study hypotheses. SEM allows concurrent testing of all hypotheses together with direct and indirect effects. The results of the direct effects show that Effort Expectancy, Facilitating Conditions, Performance Expectancy, Social Influence, Trust in SANAD App, Trust towards SANAD App, and Perceived Service Quality are positively and significantly affected Attitude; thus H1–H4, and H6, H7, and H8 were accepted; while Perceived Risk did not have influences on Attitude (β = 0.025); so, H5 was rejected. In addition, Attitude positively and significantly affected Intention to Use, and the latter on Intention to Recommend, and in turn, Public Health Protection; therefore, H9, H10, and H11 were accepted. Moreover, the coefficient of determination (R^2^) for the research endogenous variables for Attitude, Intention to Use, Intention to Recommend, and Public Health Protection were 0.485, 0.661, 0.757, and 0.709 respectively, which shows that the model does account for the variation of the proposed model. Table 6 provides a brief of the analyzed hypotheses.

### 4.3. Moderation Effects

Hypotheses H12, H13, H14, and H15 argued that there is a significant difference in the respondent Attitude due to age, education level, gender, and internet experience. Independent Samples *t*-test was employed to examine if there were any significant differences in the respondent’s Attitude that can be attributed to gender. Also, the ANOVA test was employed to examine if there are any significant differences in the respondent Attitude that can be attributed to age, education, and internet experience. Results of the *t*-test, shown in Table 7, showed that there is a significant difference in the Attitude that can be attributed to gender, that goes for females than males.

Additionally, the outcomes of the ANOVA test, displayed in Table 8, showed that there is a significant difference in the respondent’s Attitude in favor of age and education, while internet experience did not. This is to confirm the statistical significance of the differences between each pair of the five groups for age (i.e., 18 to less than 34, 34 to less than 44, 44 to less than 54, 54 to less than 64, and 64 and over), and for the differences between each pair of the five groups for education (i.e., High school and less, Diploma, Bachelor, Master, Ph.D.) were statistically different from one another.

### 4.4. Machine Learning Techniques Validation & Prediction

Previous research such as [66] used machine learning techniques for validation and prediction. The study confirms five Machine Learning (ML) classification techniques, in which the inherited knowledge in the input of a dataset is extracted to a form of desired output pattern [67]. Five ML models are used to build and evaluate models for the SANAD application, which are Artificial Neural Network (ANN) [68], Linear Regression [69], Sequential Minimal Optimization algorithm for Support Vector Machine (SMO) [70], Bagging with REFTree model [71], and Random Forest [72]. To estimate the error values between the prediction and actual output values, the ANN employs the back-propagation technique. The error is then used to update the ANN architecture’s weights and bias parameters, bringing the predicted and real values closer together. Linear regression is a polynomial function model in which the independent variables have weighted coefficients, and the output is target dependent. In a repetition of processes, the training phase updates the coefficients of the linear function from the training dataset. The SMO technique changes the weighted vectors of the SVM model using the Sequential Minimal Optimization algorithm. The bagging technique generates many REFTree models from a random sampling of the objects and attributes in the training set, and the average value of the trees provides the final predicted value. The Random Forest is a set of Decision Trees (DT) models that uses a random sample of training data objects and random attribute subsets for each sub-tree. The average value of the DT trees represents the model’s ultimate outcome.

### 4.5. Results

In this study, the inclination of customers to use SANAD applications is discussed. Several factors determine whether to use this technology or promote it to others. The most important value is to relate aspects to customer proclivity in well-formed patterns that recapitulate data from many customers. However, in a classification notion, the ML field includes tools for building models of several types that relate independent factors to dependent variables. This study confirms four relationship models, which are the effects of UTAUT, TAM & TPB, and SQ factors on attitude as a dependent variable, attitude as an independent factor to CIU variable, CIU to IRC, and URC to PHP variable. The results of five machine learning algorithms applied to four relationship models illustrated in Figure 1. The models are represented on the x-axis, while the R^2^ and Mean Square Error (MSE) values are displayed on the y-axis. The R^2^ reflects how the dependent variable (target) is projected to vary because of the independent values. The MSE indicates the average distance between the evaluated and actual output values of a model. The random forest and Bagging_REPTree ML models, as illustrated in Figure 2 of the R^2^ values to the target values, produce respectable results when compared to the other ML techniques of the four models. This shows that the tree-based models are more correlated to the target labels. Model 1’s ML, apart from the ANN model, uses an efficient prediction for each parameter integrated into the model to obtain the power to predict the attitude variable from the independent variables. The other relationship models are gradually increasing R2 values in the effective models. These findings show that implementing the SANAD application in predicting public healthcare based on people’s attitudes has an impact, with some even promoting it to others. Furthermore, Figure 3 ensures the effectiveness of the SMO model that achieves a low MSE value between the actual and the target values of the model.

## 5. Discussion

Using mobile e-government services for the purpose of public health protection is a new concept. Yet, as the need for such applications rises people are more willing to battle the notion. As the world is battling with COVID-19 public health issue are becoming the most prominent issue. SANAD App came as a solution to a problem with too many ever-changing variables. This study aimed to discover the factors that influence the use of the SANAD App And hence the effect of such factors on public health protection. The study had many contributions on both theoretical and practical levels and may help in improving the interaction between application users and the application itself. In Table 2, Effort expectancy and Facilitating conditions both had high means which means participants find the SANAD App. They are easy to use and they have accessibility to use such an application. Further, Intention to Recommend has the highest mean among the mediating factors which reflects the user’s desire to recommend such an application. Almost 49% of the questionnaire participants agreed or strongly agreed with the SANAD App Participate in behavioral changes to protect the participant and others from infection and educate others about the pandemic. On the other hand, 29% disagreed and strongly disagreed. Hence, participants believe in the role of the SANAD App In public health protection. As for intention to recommend, 45% of the participants will recommend SANAD App While 30% disagreed or strongly disagreed. Further, 41% of the participants will recommend the SANAD App To friends on social networks, while 31% disagreed or strongly disagreed. In addition, 53% of the participants will recommend the SANAD App If they had a pleasant experience with it, on the other hand, 25% disagreed or strongly disagreed. The participants showed that they intend to reuse, continue to use, and expect to use SANAD App (47%, 44%, 45%) respectively. While (30%, 33%, and 29%) respectively disagreed or strongly disagreed. Regarding Attitude, 51% of the participants thought of the SAND App as a promising idea, 48% liked accessing government service through SANAD App, and 39% believe that using SANAD App would be a pleasant experience? On the other hand (26%, 31%, and 39%) disagreed or strongly disagreed, respectively. As COVID-19 mutates and new variants appear over time, the need for such applications may be rising. Hence, Laws and regulations should accommodate and reflect such situations. From the findings listed above people are willing to tolerate some privacy issues in return for public health protection measures. Where developers of SANAD App should take into consideration people’s needs and requirements. Hence design and development during the system development life cycle should be a major concern. This study proposed a theoretical model to improve knowledge about e-government applications, service adoption, and acceptance. The theoretical model is then confirmed using SEM and ML techniques with collected data. Prior studies suggested and verified that the UTAUT, TPB, and TAM are the primary e-government adoption models. Many studies investigated varied factors to prove that it has influenced the success of e-government applications, and service adoption, and had a significant impact on the individual’s attitude to using the e-government applications and services. The current research found that from people’s perspectives, the performance expectancy provides benefits in performing specific activities and improves performance. As a result, it has a positive influence on the attitude toward SANAD services adoption which aligns with the outcomes of [6,22,23,24,66]. Moreover, the results showed that Effort Expectancy has a positive influence on Attitude toward SANAD Services Adoption which confirms the findings of [2,19,23,25]. Also, current research confirms that the Social Influence affects the Attitude towards SANAD Services Adoption where friends, close relatives, important people they respect, and others in the community inspire users’ judgments and choices toward the adoption and use of SANAD App and Services which confirms the outcomes [2,23,25] that there is a direct impact of SI on the individuals’ attitudes towards using government e-service and application. Also, our findings are consistent with the findings of [23,28,29,73] that Facilitating Conditions positively influence the Attitude towards SANAD Services Adoption where the availability of the necessary resources encourages citizens to access and use the SANAD App Services.

Meanwhile, the result shows that the perceived Risk influences the Attitude positively toward SANAD App Services Adoption which will result in affecting the intention to use. These results are not consistent with those [19,39,74], where researchers found that Perceived Risk (PR) might limit the adoption of the interaction of citizens with the government e-services and application. And as a result, citizens might have a negative impact on ATT toward reusing government e-services and applications. accordingly, e-government Apps and services providers should handle risk-reduction approaches and strategies.

In addition, the current research results show that Trust and quality of service influence the citizen’s attitude and have a significant impact on attitudes regarding SANAD service adoption, where the Trust includes Trust towards and in the SANAD App The Trust hypothesis results agree with the results of researchers who investigated and studied the aspects of trust and their argument that citizens’ loyalty to e-government services is related to [19,29,32]. Meanwhile, [3] found that the quality of service has the greatest influence on citizen loyalty. As well as the current research results confirm that attitude is positively and significantly affected the Intention to use which concurs with [19,29] results. where researchers claimed that attitudes have a direct impact on behavioral intentions to use e-government services.

Also, our research outcomes show that the Intention to use has an influenced the intention to recommend SANAD services positively, such outcomes are in agreement with the findings of the previous studies by [2,19,29,44] which confirm that users who intend to use the government e-services and application have a positive influence on intention to recommend its adoption. The recommendation of using the SANAD App has a behavioral impact and change on the Jordanian society through increasing awareness of public health prevention using technology from COVID-19. This concludes with the findings of [28] that modern technologies can positively affect the awareness and increase the public protection against the COVID-19.

Furthermore, among the mediating factors, Intention to recommend has the highest mean, reflecting the user’s willingness to recommend such an application to other people. Hypotheses H12, H13, H14, and H15 argued that there is a significant difference in the respondent’s Attitude due to gender, age, education level, and internet experience. Results showed that there is a significant difference in the Attitude that can be attributed to gender for females than males. The findings also show that there is a significant difference in the respondent Attitude in favor of age and education, while internet experience did not. Indeed, there were statistical differences among the five age groups, and the five educational groups were statistically different from one another.

### 5.1. Theoretical Contributions

This research aimed to study varied factors: independent, mediating, and moderators that influence the e-government SANAD App as an e-government public health protection tool. Many studies were conducted in this regard, as shown in the references section, still only this research used such a model to study the e-government service application SANAD App for the goal of public health protection. This study will enrich the literature regarding specially developed mobile applications used to protect the public health in this COVID-19 pandemic where the study discovered the influence of eight independent factors, with four moderating factors on attitudes as intermediate factors, and intention to use, intention to recommend, and public health protection. Also, the research studied service quality in-depth by including all 5 dimensions Tangible, reliability, responsiveness, assurance, and empathy. Moreover, the trust factors with two perspectives: trust in SANAD App and towards SANAD App In addition, the research included one moderating factor that is unique to such research internet experience.

### 5.2. Practical Implications

Mobile applications in e-government are a necessity, especially with the COVID-19 pandemic and the never-ending mutations of COVID-19. The government of Jordan will need to issue new laws and regulations about the use of mobile e-government applications. Laws must be current and Up to Date to keep up with the technology and the needs for such technology and users. the governments and legislative should be able to respond to different sociological, economic, health, and political changes on both national and international levels. As the need is the mother of invention, governments, and legal organizations, must learn that germs and diseases know no border, hence the legal body must identify with the need for mobile e-government services. Mobile e-government services will save much-needed resources like time and money but will also save lives, hence the need for such applications must be appreciated. Hence, such applications must be well designed and developed by highly skilled professionals and not be left immature.

Governments must be able to cooperate to create mobile applications that interact with each other. Different countries are issuing applications much like SANAD App, hence such applications should be compatible with each other to help and ease the users. Moderating factors age, gender, and educational level did have an influence on the attitude of the participants. Younger, female, and highly educated participants had a more positive attitude towards the use of the SANAD App Hence, the government should enable older and less educated people by involving and training such a population. Many venues can be used for such endeavors. Further, other moderating factors (and internet experience) had no influence on the attitude towards the use of the SANAD App As such, a niche was shown for the government to use when designing and developing such applications when the need arises.

### 5.3. Academic Implications

Based on the results and findings of this study. It provides a significant contribution to the literature on evaluating and predicting the effectiveness of e-government services. The study model was developed using different models including TAM, TPB, and UTAUT in addition to Risk and Trust. Furthermore, the outcome shows that the proposed model is stable with significant interpretability, which contributes to the studies of e-government services adoption. Future studies can use and extend the proposed model to improve the investigation into the adoption of e-government apps and services, commercial e-services, and other public and social e-services.

### 5.4. Limitations and Future Research

As the study was implemented in one of many COVID-19 flares, access to participants was not easy. As such, limiting the researchers’ access to participants and the ease with which interviews are to be conducted, many venues were used to access a better number of participants i.e., WhatsApp groups, Facebook, etc. Also, the researchers only collected 442 Reponses. This work is limited to SANAD App Only offered in Jordan, future work could be proposed to compare with other countries and comparing results with diverse cultures. Also, the results showed that 90% of the participants are young with a bachelor’s degree and good experience as Internet users, future research must be conducted to study the older age people’s attitude and the effect of such technologies on their health and public protection. In addition, a more in-depth study can be conducted where participants are less familiar with the use of mobile technology.

## 6. Conclusions

In conclusion, this research investigates the varied factors that influence the use of smartphone applications in e-government for the purpose of health protection during the COVID-19 pandemic. The proposed model is developed using UTAUT, TPB, and TAM models in addition to, trust and perceived risk constructs. The collected data were analyzed using structural equation modeling (SEM), (CFA), and (ML) methods. The results showed that almost 49% of the questionnaire’s participants agreed or strongly agreed that SANAD App Affected and changed their behavior and acted in protecting the participants and others from COVID-19 19 infection and in educating others about the pandemic. According to respondents’ responses, the following constructs; performance expectancy, effort expectancy, social influence, facilitating conditions, perceived risk, trust, and perceived service quality were significant antecedents for participants’ attitudes to adopt and use the e-government SANAD App Services, where the strength of these relationships is affected by the moderating variables, including age, gender, educational level, and internet experience on behavioral intention. On the other hand, the perceived risk did not have a significant effect on attitude towards SANAD App Adoption. Also, the results showed that attitude positively and significantly affected SANAD App intention to use, and the latter on intention to recommend the App to others which increased the awareness of public health protection. Consequently, we can conclude the following interesting findings; first: the results showed that highly educated participants had a more positive attitude toward the use of the SANAD App Thus, the government should make it easier for older and less educated individuals to accept new technology by educating and teaching them. The second, using SANAD App has a behavioral impact on the Jordanian society through increasing awareness of public health protection against the COVID-19.

## Figures and Tables

**Figure 1 ijerph-19-08281-f001:**
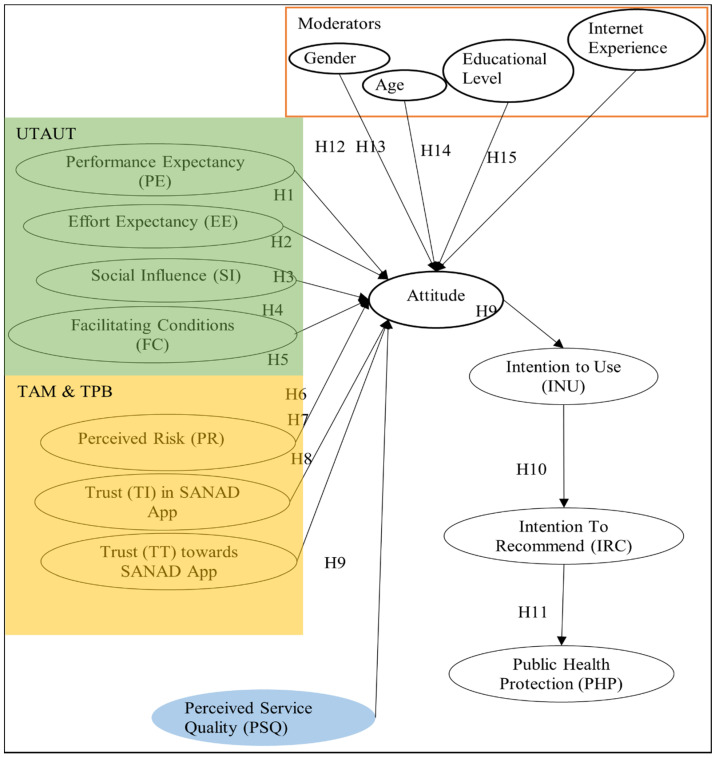
A suggested model of the research.

**Figure 2 ijerph-19-08281-f002:**
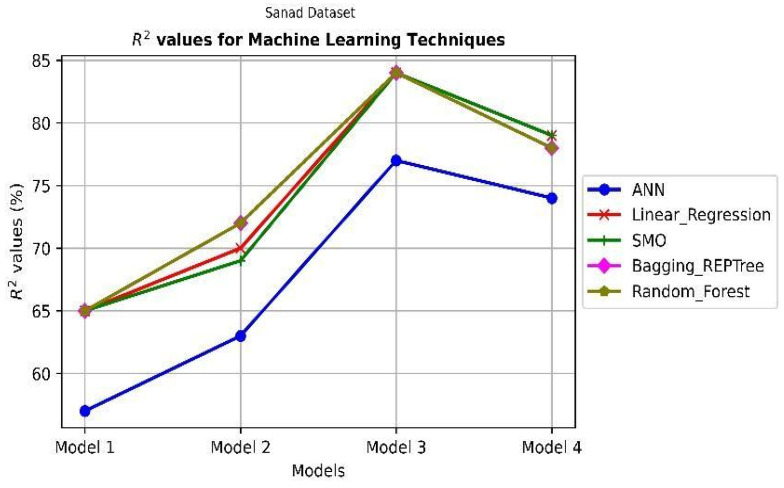
The results of using ML techniques on the SANAD App dataset R^2^.

**Figure 3 ijerph-19-08281-f003:**
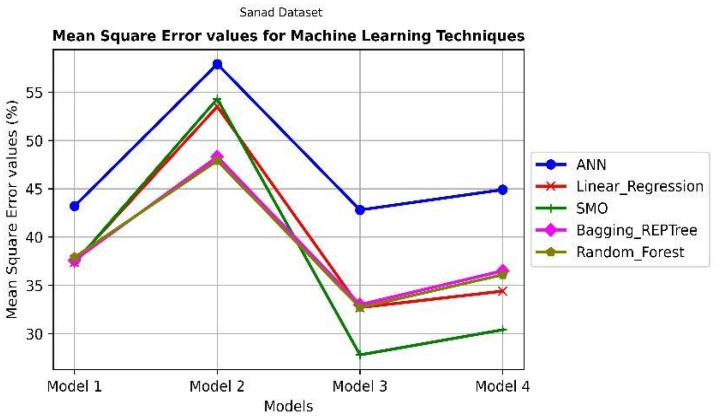
The results of using ML techniques on SANAD App MSE.

**Table 1 ijerph-19-08281-t001:** Description of the respondents’ demographic profiles.

Category	Category	Frequency	Percentage %
Gender	Male	186	42.1
	Female	256	57.9
	Total	442	100
Age (Year)	18 to less than 34	398	90.04
	34 to less than 44	24	5.43
	44 to less than 54	11	2.49
	54 to less than 64	9	2.04
	Total	442	100
Education level	High school and less	14	3.17
	Diploma	13	2.94
	Bachelor	387	87.56
	Master	10	2.26
	PhD	18	4.07
	Total	442	100
Internet experience	Low	13	2.9
	Good	194	43.9
	Excellent	235	53.2
	Total	442	100

**Table 2 ijerph-19-08281-t002:** Overall mean and standard deviation of the study’s variables.

Type of Variable	Variables	Mean	Standard Deviation	Level	Order
	Performance Expectancy (PE)	3.2511	1.09996	Moderate	3
	Effort Expectancy (EE)	3.4661	1.03303	High	2
	Social Influence (SI)	3.2149	1.09710	Moderate	7
	Facilitating Conditions (FC)	3.5830	1.02785	High	1
	Perceived Risk (PR)	3.0649	1.09711	Moderate	8
	Trust in SANAD App (TI)	3.3069	1.15473	Moderate	4
	Trust towards SANAD App (TT)	3.2971	1.17361	Moderate	5
	Perceived Service Quality (PSQ)	3.2464	1.04962	Moderate	6
Mediating Variable	Attitude (ATT)	3.1816	0.84498	Moderate	2
	Intention to Use (INU)	3.1448	1.22958	Moderate	3
	Intention to Recommend (IRC)	3.2187	1.16445	Moderate	1
Dependent Variable	Public Health Protection (PHP)	3.2481	1.20824	Moderate	-

**Table 3 ijerph-19-08281-t003:** Mean and standard deviation of the study’s variables.

Performance Expectancy (PE)	Mean	SD	Level	Order
PE1	3.30	1.134	Moderate	1
PE2	3.26	1.154	Moderate	2
PE3	3.19	1.157	Moderate	3
Effort Expectancy (EE)	Mean	SD	Level	Order
EE1	3.51	1.117	High	1
EE2	3.45	1.136	High	3
EE3	3.40	1.098	Moderate	4
EE4	3.50	1.107	High	2
Social Influence (SI)	Mean	SD	Level	Order
SI1	3.19	1.191	Moderate	2
SI2	3.28	1.138	Moderate	1
SI3	3.17	1.193	Moderate	3
Facilitating Conditions (FC)	Mean	SD	Level	Order
FC1	3.82	1.082	High	1
FC2	3.61	1.168	High	2
FC3	3.32	1.223	Moderate	3
Perceived Risk (PR)	Mean	SD	Level	Order
PR1	3.05	1.195	Moderate	2
PR2	3.15	1.172	Moderate	1
PR3	3.00	1.170	Moderate	3
Trust in SANAD App (TI)	Mean	SD	Level	Order
TI1	3.29	1.178	Moderate	2
TI2	3.28	1.229	Moderate	3
TI3	3.34	1.184	Very high	1
Trust towards SANAD App (TT)	Mean	SD	Level	Order
TT1	3.32	1.206	Moderate	1
TT2	3.29	1.190	Moderate	2
TT3	3.29	1.223	Moderate	2
Perceived Service Quality (PSQ)	Mean	SD	Level	Order
Service Quality Tangibles (SQT)				
PS1	3.31	1.152	Moderate	1
PS2	3.27	1.150	Moderate	3
PS3	3.28	1.169	Moderate	2
PS4	3.24	1.166	Moderate	4
Service Quality Reliability (SQR)				
PS5	3.30	1.142	Moderate	1
PS6	3.12	1.201	Moderate	4
PS7	3.20	1.196	Moderate	3
PS8	3.24	1.165	Moderate	2
Service Quality Responsiveness (SQP)				
PS9	3.23	1.143	Moderate	2
PS10	3.26	1.154	Moderate	1
PS11	3.18	1.137	Moderate	4
PS12	3.21	1.153	Moderate	3
Service Quality Assurance (SQA)				
PS13	3.24	1.156	Moderate	2
PS14	3.28	1.132	Moderate	1
PS15	3.23	1.123	Moderate	3
Service Quality Empathy (SQE)				
PS16	3.29	1.113	Moderate	2
PS17	3.31	1.121	Moderate	1
PS18	3.23	1.159	Moderate	3
Attitude (AT)	Mean	SD	Level	Order
ATT1	3.31	1.264	Moderate	1
ATT2	3.10	1.242	Moderate	3
ATT3	3.25	1.283	Moderate	2
ATT4	3.07	1.246	Moderate	4
Intention to Use (IU)	Mean	SD	Level	Order
INU1	3.17	1.272	Moderate	1
INU2	3.11	1.286	Moderate	3
INU3	3.15	1.253	Moderate	2
Intention to Recommend (IR)	Mean	SD	Level	Order
IRC1	3.16	1.284	Moderate	2
IRC2	3.08	1.285	Moderate	3
IRC3	3.41	1.162	High	1
Public Health Protection (PHP)	Mean	SD	Level	Order
PHP1	3.26	1.260	Moderate	1
PHP2	3.26	1.223	Moderate	1
PHP3	3.23	1.234	Moderate	2

**Table 4 ijerph-19-08281-t004:** Properties of the final measurement model.

Constructs and Indicators	Factor Loadings	Std. Error	Square Multiple Correlation	Error Variance	Cronbach Alpha	Composite Reliability *	AVE **
Performance Expectancy (PE)					0.955	0.94	0.95
PE1	0.945	***	0.893	0.137			
PE2	0.951	0.024	0.904	0.128			
PE3	0.916	0.027	0.839	0.215			
Effort Expectancy (EE)					0.945	0.93	0.94
EE1	0.863	***	0.744	0.319			
EE2	0.923	0.039	0.853	0.190			
EE3	0.917	0.037	0.842	0.191			
EE4	0.896	0.039	0.803	0.241			
Social Influence (SI)					0.927	0.90	0.76
SI1	0.930	***	0.865	0.192			
SI2	0.914	0.028	0.836	0.212			
SI3	0.855	0.033	0.731	0.383			
Facilitating Conditions (FC)					0.864	0.83	0.87
FC1	0.784	***	0.614	0.451			
FC2	0.916	0.062	0.838	0.220			
FC3	0.805	0.064	0.648	0.525			
Perceived Risk (PR)					0.922	0.89	0.91
PR1	0.885	***	0.784	0.308			
PR2	0.915	0.037	0.837	0.223			
PR3	0.882	0.038	0.777	0.305			
Trust in SANAD App (TI)					0.963	0.94	0.96
TI1	0.957	***	0.916	0.116			
TI2	0.941	0.023	0.885	0.174			
TI3	0.942	0.022	0.888	0.157			
Trust towards SANAD App (TT)					0.971	0.95	0.88
TT1	0.950	***	0.902	0.142			
TT2	0.954	0.022	0.910	0.127			
TT3	0.969	0.021	0.939	0.091			
Perceived Service Quality (PSQ)					0.988	0.98	0.98
PSQ1	0.815	***	0.665	0.444			
PSQ2	0.891	0.046	0.793	0.273			
PSQ3	0.888	0.047	0.788	0.289			
PSQ4	0.905	0.046	0.819	0.245			
PSQ5	0.925	0.044	0.855	0.189			
PSQ6	0.913	0.047	0.833	0.240			
PSQ7	0.914	0.047	0.835	0.235			
PSQ8	0.911	0.046	0.831	0.229			
PSQ9	0.916	0.045	0.839	0.210			
PSQ10	0.931	0.044	0.867	0.177			
PSQ11	0.929	0.044	0.864	0.176			
PSQ12	0.929	0.045	0.863	0.182			
PSQ13	0.918	0.045	0.843	0.209			
PSQ14	0.898	0.045	0.806	0.247			
PSQ15	0.913	0.044	0.834	0.209			
PSQ16	0.916	0.043	0.839	0.199			
PSQ17	0.907	0.044	0.824	0.221			
PSQ18	0.888	0.046	0.788	0.284			
Attitude (ATT)					0.910	0.86	0.68
ATT1	0.834	***	0.695	0.486			
ATT3	0.884	0.044	0.781	0.360			
ATT4	0.915	0.042	0.836	0.254			
Intention to Use (INU)					0.966	0.94	0.95
INU1	0.957	***	0.917	0.134			
INU2	0.962	0.020	0.926	0.122			
INU3	0.934	0.023	0.873	0.199			
Intention to Recommend (IRC)					0.928	0.90	0.92
IRC1	0.968	***	0.936	0.105			
IRC2	0.957	0.019	0.916	0.138			
IRC3	0.793	0.029	0.629	0.500			
Public Health Protection (PHP)					0.974	0.96	0.97
PHP1	0.953	***	0.908	0.146			
PHP2	0.965	0.020	0.930	0.104			
PHP3	0.970	0.020	0.941	0.090			

* Employing [65] formula of the composite reliability & ** The formula for the variance extracted is: Average Variance Extracted = Σ Li^2^/(Σ Li^2^ + Σ Var (Ei)) where Li is the standardized factor loadings for each indicator, and Var (Ei) is the error variance associated with the individual indicator variables. *** null value.

**Table 5 ijerph-19-08281-t005:** Correlations of constructs.

Constructs	PE	EE	SI	FC	PR	TI	TT	PSQ	ATT	INU	IRC	PHP
PE	0.97											
EE	0.871	0.96										
SI	0.932	0.822	0.87									
FC	0.625	0.742	0.660	0.93								
PR	0.375	0.364	0.361	0.296	0.95							
TI	0.882	0.837	0.862	0.577	0.274	0.97						
TT	0.861	0.809	0.853	0.543	0.274	0.967	0.93					
PSQ	0.941	0.868	0.810	0.594	0.391	0.917	0.892	0.98				
ATT	0.859	0.755	0.850	0.508	0.283	0.889	0.894	0.877	0.82			
INU	0.831	0.766	0.835	0.541	0.258	0.883	0.891	0.850	0.959	0.97		
IRC	0.821	0.750	0.833	0.518	0.254	0.884	0.892	0.851	0.953	0.959	0.96	
PHP	0.848	0.757	0.869	0.543	0.298	0.902	0.909	0.879	0.899	0.879	0.909	0.98

Note: Diagonal elements are square roots of the average variance extracted for each of the ten constructs. Off-diagonal elements are the correlations between constructs.

**Table 6 ijerph-19-08281-t006:** Summary of proposed results for the theoretical model.

Research Proposed Paths	Coefficient Value	*t*-Value	*p*-Value	Empirical Evidence
H1: PE → ATT	0.180	7.239	0.000	Supported
H2: EE → ATT	0.063	2.393	0.017	Supported
H3: SI → ATT	0.113	4.548	0.000	Supported
H4: FC → ATT	0.055	2.071	0.038	Supported
H5: PR → ATT	0.025	0.997	0.319	Not supported
H6: TI → ATT	0.193	8.130	0.000	Supported
H7: TT → ATT	0.307	13.169	0.000	Supported
H8: PSQ → ATT	0.249	9.546	0.000	Supported
H9: ATT → INU	0.948	29.328	0.000	Supported
H9: INU → IRC	0.870	37.035	0.000	Supported
H9: IRC → PHP	0.923	32.760	0.000	Supported

**Table 7 ijerph-19-08281-t007:** *t*-test of the respondent Attitude attributed to gender.

Variable	Male			Female			T	df	Sig.
	N	Mean	Std. Dev.	N	Mean	Std. Dev.			
Attitude	186	3.0179	1.25764	256	3.3503	1.0717	2.916	359.196	0.004

**Table 8 ijerph-19-08281-t008:** ANOVA Analysis of respondent Attitude attributed to Age, Educational Level, and Internet Experience.

Sig.	F	Mean Square	Df	Sum of Squares		Variable
0.011	3.766	5.008	3	15.023	Between Groups	Attitude attribute to age
		1.330	438	582.409	Within Groups	
			441	597.432	Total	
0.042	2.502	3.345	4	13.378	Between Groups	Attitude attribute to education
		1.337	437	584.054	Within Groups	
			441	597.432	Total	
0.652	0.429	0.582	2	1.165	Between Groups	Attitude attribute to internet experience
		1.358	439	596.267	Within Groups	
			441	597.432	Total	

## Data Availability

Not applicable.

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
