# Peer review of "Predictors for E-Government Adoption of SANAD App Services Integrating UTAUT, TPB, TAM, Trust, and Perceived Risk"

_ijerph, 2022, doi:10.3390/ijerph19148281_

Round 1
Reviewer 1 Report
The framework presented by the authors is based on more than 60 studies. It would be important for the authors to inform the reader why they selected these studies and not others, that is, to explain how the selection process of these studies was.
It is also important to show the questionnaire used in this study (authors can show as an appendice, for example).
Explain how the 14 variables were arrived at (there were 15 hypotheses and the first relationship that the reader can make is a variable for each hypothesis).
The profile of the participants can affect the result. 90% are young, with a bachelor's degree and good experience as Internet users, this is an audience that is more open to using technologies. Other participant profiles could lead the search to a different result (it is important to mention this in the study).
I didn't check the stats.
Author Response
Dear Reviewer (1)
We appreciate the time and effort that you dedicated to providing feedback on our manuscript and we are grateful for the insightful & valuable comments that resulted in significant improvements to the manuscript. We have incorporated all the suggestions made by the reviewers. Those changes can be tracked within the manuscript.
Please see below, in bold, a point-by-point response to your comments and suggestions.
Thank you
Comments and Suggestions for Authors
1- The framework presented by the authors is based on more than 60 studies. It would be important for the authors to inform the reader why they selected these studies and not others, that is, to explain how the selection process of these studies was.
The authors select the most current papers related to e-government service (17 papers). (12 of them are after 2018).
- Quality 9
- ML : 7
- UTAUT (17)
- TAM (9)
- PBT (4)
To support the literature of this study, the researchers depended on current published research that pertains to e-government services and quality almost referenced (26) papers on this topic. Each hypothesis was referenced by at least 5-6 current published papers. Each used model was referenced by UTAUT 17 studies, TAM 9 studies, PBT 4 studies.
Please see the end of section 2
2- It is also important to show the questionnaire used in this study (authors can show as an appendice, for example).
The study questionnaire has been added as an appendix (A1)
3- Explain how the 14 variables were arrived at (there were 15 hypotheses and the first relationship that the reader can make is a variable for each hypothesis).
The researchers wanted to study in detail the variable hence, to concentrate the efforts almost every variable was set in one hypothesis
4- The profile of the participants can affect the result. 90% are young, with a bachelor's degree and good experience as Internet users, this is an audience that is more open to using technologies. Other participant profiles could lead the search to a different result (it is important to mention this in the study).
1According to United Nations projections Jordan's literacy rate for 2018 was 98.23%
2Also, the median age in Jordan is 23.8 years, which means half of the population is below that age. Hence the population of the study is really a reflection of the Jordanian population.
Also mentioned in challenges and future work
1https://www.macrotrends.net/countries/JOR/jordan/literacy-rate
2https://www.worldometers.info/world-population/jordan-population/
Thank you
Reviewer 2 Report
Research summary In this manuscript, the authors present have adapted various measuring items from the unified theory of acceptance and use of technology, the theory of acceptance model, and TPB expanded by perceived service quality and moderating factors (age, gender, educational level, and Internet experience).
Major Strengths: The major strengths of the research are:
- The construction of a new model of Integrating UTAUT, TPB, TAM Trust, and Perceived Risk.
- In addition, public health services should continue their efforts to improve factors that influence attitudes toward such mobile services applications; as well as securing the accessibility of e-government services.
Major Weaknesses: The major weaknesses of the research are:
- Lack of quality of figure 1. A suggested model of the research
- Authors should detail of H2: Effort Expectancy (EE) has a positive influence related to Attitude (ATT) towards SANAD Services Adoption:
- H3: Social Influence (SI) has a positive influence related to Attitude (ATT) towards SANAD Services Adoption.
- H6: Trust towards SANAD App. (TT) has a positive influence related to Attitude (ATT) towards SANAD App. Services Adoption. Trust is related to the belief in accepting the conditions of use and rules of the app when accessing user data sources. Authors should expand the argument of H6. It needs minor revisions.
- In addition, Authors should a greater technological contribution.
- - It is not very clear study’s variables (e.g. Performance Expectancy (PE) - PE1 ? PE2 ? PE3? )
- It is not very clear of Practical implications and Limitations and future research.
The authors need to become familiar with up-to-date TAM, TPB, and UTAUT in addition to Risk and Trust. methods. They also need to present some baseline methods and show case why their methodology performs better.
It is not clear why the authors selected to represent the TAM, TPB, and UTAUT in addition to Risk and Trust.
H5: Perceived Risk (PR) has a Negative influence related to Attitude (ATT) towards SANAD Services Adoption.
H5: PR → ATT Not supported. In this context, it would be necessary for the paper to explain why not supported.
Grammar and Readability: The manuscript is perfectly readable.
Specific Comments: The work is really interesting and can be a great scientific contribution. However, there are large gaps when building the corpus related performance expectancy, effort expectancy, social influence, easing conditions,
risk and trust, and service quality affect the attitude towards SANAD App.
Concluding Remarks: In short, the original idea of the paper, that is, the new corpus on the stance to the literature on evaluating and predicting the effectiveness of e-government services. The study model was developed using different models including TAM, TPB, and UTAUT in addition to Risk and Trust.
Author Response
Dear Reviewer (2)
We appreciate the time and effort that you dedicated to providing feedback on our manuscript and we are grateful for the insightful & valuable comments that resulted in significant improvements to the manuscript. We have incorporated all the suggestions made by the reviewers. Those changes can be tracked within the manuscript.
Please see below, in bold, a point-by-point response to your comments and suggestions.
Thank you
Comments and Suggestions for Authors
Research summary In this manuscript, the authors present have adapted various measuring items from the unified theory of acceptance and use of technology, the theory of acceptance model, and TPB expanded by perceived service quality and moderating factors (age, gender, educational level, and Internet experience).
Major Strengths: The major strengths of the research are:
- In addition, public health services should continue their efforts to improve factors that influence attitudes toward such mobile services applications; as well as securing the accessibility of e-government services.
Thank you
Major Weaknesses: The major weaknesses of the research are:
1- Lack of quality of figure 1. A suggested model of the research
We apologize for the quality, rectified please see the new Figure 1
2 - Authors should detail of H2: Effort Expectancy (EE) has a positive influence related to Attitude (ATT) towards SANAD Services Adoption:
Amended as per requested, please refer to H2 in paper
3 - H3: Social Influence (SI) has a positive influence related to Attitude (ATT) towards SANAD Services Adoption.
Amended as per requested, please refer to H3 in paper
4 - H6: Trust towards SANAD App. (TT) has a positive influence related to Attitude (ATT) towards SANAD App. Services Adoption. Trust is related to the belief in accepting the conditions of use and rules of the app when accessing user data sources. Authors should expand the argument of H6. It needs minor revisions.
Amended as per requested, please refer to H6 in paper
5 - In addition, Authors should a greater technological contribution.
- - It is not very clear study’s variables (e.g. Performance Expectancy (PE) - PE1 ? PE2 ? PE3? )
The study questionnaire has been added as an appendix (A1) and you can find all the study’s variables and items.
6 - It is not very clear of Practical implications and Limitations and future research.
please refer to the "Discussion" section, you can find: Theoretical contributions, Practical implications, Academic implications and Limitations, and future research
7- The authors need to become familiar with up-to-date TAM, TPB, and UTAUT in addition to Risk and Trust. methods. They also need to present some baseline methods and show case why their methodology performs better.
It is not clear why the authors selected to represent the TAM, TPB, and UTAUT in addition to Risk and Trust.
The factors were not studied in this wholesome idea anywhere
8- H5: Perceived Risk (PR) has a Negative influence related to Attitude (ATT) towards SANAD Services Adoption.
H5: PR → ATT Not supported. In this context, it would be necessary for the paper to explain why not supported.
PR ranked the lowest among constructs and the mean was 3.0649 as per table 2. The items' values of PR1, PR2, PR3 were (3.05,3.15,3.00) as per table 3. Reflects respondents are neutral when asked about their feeling about the risk related to SANAD App. The respondents’ feelings are around the term “Neutral” or “Undecided” which is intuitively understandable since the use of SANAD App. is new to the public.
A discussion was added in the pertaining section.
9- Grammar and Readability: The manuscript is perfectly readable.
Thank you
Specific Comments: The work is really interesting and can be a great scientific contribution. However, there are large gaps when building the corpus related performance expectancy, effort expectancy, social influence, easing conditions, risk and trust, and service quality affect the attitude towards SANAD App.
Concluding Remarks: In short, the original idea of the paper, that is, the new corpus on the stance to the literature on evaluating and predicting the effectiveness of e-government services. The study model was developed using different models including TAM, TPB, and UTAUT in addition to Risk and Trust.
Thank you
Reviewer 3 Report
Review of Predictors for E-government Adoption of SANAD App. 2 Services Integrating UTAUT, TPB, TAM Trust, and Perceived 3 Risk. Submitted to International Journal of Environmental Research and Public Health.
The study is interested in understanding the adoption of mobile applications for e-government or e-health. The measurements from the study are clearly described and referred to suitably. Besides that, I see several flaws with the article and therefore add the grade of a significant revision to it. The problem is the detailed level of what is written besides the study’s presentation. The detail level is either too high, implying that the reader does not understand why the text is there or too detailed, meaning that readers cannot sort out the details. I will point out some of the flaws.
- You discuss terms of e-government when it some to this app. I am curious about why e-government and not e-health, mainly since you have submitted to a journal in e-health. E-government is broad, and I think you should be more detailed about your chosen field.
- The abstract is far too extended, and I get lost in it. At its beginning, I would like to have one sentence about why this study is interesting. Right now, I am just thrown into many factors. Then the factors are brought up in detail, which is far too detailed information. Describe that you have used specific models, the research design, the results, and the implications.
3.
- The introduction is hard to understand, especially regarding why this study is done. You have one or two sentences on why, on a very high level, and then comes a literature review. This review is hard to grab and why I need to read it. Especially when it comes to literature written thirty years ago. Skip the literature review and argue for your study after you have settled your research area for this article. One suggestion is to use the CARS model; see https://libguides.usc.edu/writingguide/CARS.
- Argue for why this study is of interest. I do not understand why it is as the introduction is written today.
- 3.1 Research context could be much more interesting than it is today. At least skip the sentence: “As said, all is fair in love and war.”
- What is your population?
- Adding the ML techniques and validation is exciting but comes out of the blue for the reader. Why was that not introduced earlier?
- The discussion is either too detailed or too general level. One example is the second paragraph, where I do not understand your conclusions. Why is it so that laws need to be modified? Another example is the paragraph (or maybe you should call it a section) starting with: Using mobile e-government services for… You need to look into its structure of it, its length and also its contributions. What do you want to say about it?
Author Response
Dear Reviewer (3)
We appreciate the time and effort that you dedicated to providing feedback on our manuscript and we are grateful for the insightful & valuable comments that resulted in significant improvements to the manuscript. We have incorporated all the suggestions made by the reviewers. Those changes can be tracked within the manuscript.
Please see below, in bold, a point-by-point response to your comments and suggestions.
Thank you
Comments and Suggestions for Authors
Review of Predictors for E-government Adoption of SANAD App. 2 Services Integrating UTAUT, TPB, TAM Trust, and Perceived 3 Risk. Submitted to International Journal of Environmental Research and Public Health.
The study is interested in understanding the adoption of mobile applications for e-government or e-health. The measurements from the study are clearly described and referred to suitably. Besides that, I see several flaws with the article and therefore add the grade of a significant revision to it. The problem is the detailed level of what is written besides the study’s presentation. The detail level is either too high, implying that the reader does not understand why the text is there or too detailed, meaning that readers cannot sort out the details. I will point out some of the flaws.
- you discuss terms of e-government when it some to this app. I am curious about why e-government and not e-health, mainly since you have submitted to a journal in e-health. E-government is broad, and I think you should be more detailed about your chosen field.
The government issue this App through the ministry of IT, not The Health Ministry, and the App. Is included with other services of the government that has nothing to do with health. The App is used as a permit to enter any public facility.
- The abstract is far too extended, and I get lost in it. At its beginning, I would like to have one sentence about why this study is interesting. Right now, I am just thrown into many factors. Then the factors are brought up in detail, which is far too detailed information. Describe that you have used specific models, the research design, the results, and the implications.
Re-written as recommended
- The introduction is hard to understand, especially regarding why this study is done. You have one or two sentences on why, on a very high level, and then comes a literature review. This review is hard to grab and why I need to read it. Especially when it comes to literature written thirty years Skip the literature review and argue for your study after you have settled your research area for this article. One suggestion is to use the CARS model; see https://libguides.usc.edu/writingguide/CARS. Argue for why this study is of interest. I do not understand why it is as the introduction is written today.
Rectified, please see the added paragraphs
- Research context could be much more interesting than it is today. At least skip the sentence: “As said, all is fair in love and war.”
Rectified , please see the changes in the section
- What is your population?
SANAD App. Was requested by the government as an access permit to any public place starting from supermarkets to universities. Hence any resident in Jordan was required to install the SANAD App. On their smartphone. The population was not counted hence the authors resorted to the unknown population formula and the recommended number was 385 samples of convenience. Accordingly, the questionnaire was distributed by social media to people with the need to access public places, and 442 samples of the convenience of people living in Jordan were collected.
- Adding the ML techniques and validation is exciting but comes out of the blue for the reader. Why was that not introduced earlier?
Rectified, kindly notice changes made ( all changes are track changes)
- The discussion is either too detailed or too general level. One example is the second paragraph, where I do not understand your conclusions. Why is it so that laws need to be modified? Another example is the paragraph (or maybe you should call it a section) starting with: Using mobile e-government services for… You need to look into its structure of it, its length, and also its contributions. What do you want to say about it?
Rectified, please notice amendments done and track changes show the recommended changes.
Round 2
Reviewer 3 Report
Thanks for revising the article.
At the end of the introduction, you have several research questions coming up without any specific relevance to the previous text. I want you to consider one research question and make it more related to the introduction.
Author Response
Dear Editor,
We appreciate the time and effort that you dedicated to providing feedback on our manuscript and we are grateful for the insightful & valuable comments that resulted in significant improvements to the manuscript. We have incorporated all the suggestions made by the reviewers. Those changes can be tracked within the manuscript.
Please see below, in red, a point-by-point response to the reviewer’s (3) comments and concerns
Corresponding Author
Response:
modified in the introduction section according to your comment, and the research question is :
What are factors that influence people to use such an application when considering public health protection during the COVID-19 pandemic?